# Anti-Mycoplasma Activity of Daptomycin and Its Use for Mycoplasma Elimination in Cell Cultures of Rickettsiae

**DOI:** 10.3390/antibiotics8030123

**Published:** 2019-08-21

**Authors:** Wiwit Tantibhedhyangkul, Ekkarat Wongsawat, Sutthicha Matamnan, Naharuthai Inthasin, Jintapa Sueasuay, Yupin Suputtamongkol

**Affiliations:** 1Department of Immunology, Faculty of Medicine Siriraj Hospital, Mahidol University, Bangkok 10700, Thailand; 2Division of Infectious Diseases and Tropical Medicine, Department of Internal Medicine, Faculty of Medicine Siriraj Hospital, Mahidol University, Bangkok 10700, Thailand

**Keywords:** daptomycin, mycoplasma, cell culture, *Rickettsia*, *Orientia tsutsugamushi*

## Abstract

Mycoplasma contamination detrimentally affects cellular functions and the growth of intracellular pathogens in cell cultures. Although several mycoplasmacidal agents are commercially available for sterile cell cultures, they are not applicable to rickettsia-infected cells. In our attempt to find an anti-mycoplasma drug for contaminated rickettsial cultures, we determined the susceptibilities of three common *Mycoplasma* species to daptomycin. *Mycoplasma orale* and *M. arginini* showed low-level resistance to daptomycin (minimum inhibitory concentration, MIC = 2 mg/L), whereas *M. hyorhinis* was high-level resistant (MIC = 32 mg/L). However, some *Mycoplasma* isolates developed higher resistance to daptomycin after failed treatments with inadequate doses or durations. An aminoglycoside (gentamicin) was still active against *M. hyorhinis* and could be used in *Orientia* cultures. For complete eradication of mycoplasmas in *Rickettsia* cultures, we recommend a 3-week treatment with daptomycin at 256 mg/L. In contaminated *Orientia* cultures, daptomycin at 32 mg/L was effective in eradicating *M. orale*, whereas either gentamicin or amikacin (100 mg/L) was effective in eradicating *M. hyorhinis*. Unlike each drug alone, the combinations of daptomycin plus clindamycin and/or quinupristin/dalfopristin proved effective in eradicating *M. hyorhinis*. In summary, our study demonstrated the in vitro anti-mycoplasma activity of daptomycin and its application as a new mycoplasma decontamination method for *Rickettsia* and *Orientia* cultures.

## 1. Introduction

Mycoplasma organisms are a serious threat during cell culture because they grow well extracellularly in vitro, contaminate cell lines for a long time without being recognized, and easily spread to other cell lines. It has been reported that approximately 15–35% of continuous cell lines are contaminated by mycoplasmas [1]. Despite the absence of a cell wall, mycoplasmas are classified as Gram-positive bacteria in the class Mollicutes [2]. The common Mollicutes species causing cell culture contamination include *M*. *orale*, *M*. *arginini*, *M*. *hyorhinis*, *M*. *hominis*, *M*. *fermentans*, and *Acholeplasma laidlawii* [3]. Mycoplasma contamination may not affect the growth of cell lines, but it usually interferes with the cellular response [1] and inhibits the growth of intracellular pathogens, including *Rickettsia* and *Orientia* [4].

Mycoplasma organisms are the smallest bacteria with the smallest genomes [2]. Currently, more than 100 *Mycoplasma* species have been identified, some of which are pathogenic in humans or animals [5]. The best-known pathogenic mycoplasmas in humans are *M. pneumoniae* and *M. genitalium*. The former is a common cause of community-acquired pneumonia, but it can cause extra-pulmonary diseases as well [6]. On the other hand, *M. genitalium* is an emerging causative agent of sexually transmitted infections [7]. The two species are closely related to each other and are classified in the same pneumoniae cluster, which is distinct from other *Mycoplasma* species [8,9]. Both of these two species are often resistant to lincosamides, but intrinsically susceptible to macrolides [10]. However, acquired macrolide resistance is a growing concern [7,11]. On the contrary, mycoplasmas in other clusters are susceptible to lincosamides, but resistant to macrolides [2]. In addition to contaminating cell cultures, some *Mycoplasma* species are associated with diseases in humans or animals. *M. hominis* usually causes genitourinary tract infections; it can be an opportunistic pathogen causing disseminated diseases in immunocompromised hosts [12]. *M. hyorhinis* colonizes the nasal cavity and can cause diseases in pigs. *M. orale*, *M. fermentans* (in humans), and *M. arginini* (in animals) are commensal bacteria in the oral cavity [1]; however, they rarely cause opportunistic infections [13,14]. All these mycoplasmas are uniformly susceptible to tetracycline antibiotics [2], which are not applicable to cell cultures with rickettsiae.

Several antibiotics can get rid of mycoplasma organisms in cell culture, but these drugs may simultaneously kill rickettsial organisms. Lincosamides are a drug of choice for common mycoplasma species in cell culture [2,15] and have no effect on many Gram-negative bacteria. However, this class of antibiotics contains bacteriostatic agents, which inhibit but may not kill the bacteria. Although a previous study has demonstrated that lincomycin is effective for mycoplasma elimination in *Orientia* cultures, only two mycoplasma species (*M*. *hominis* and *M*. *orale*) were tested in that study [16]. Fluoroquinolones and aminoglycosides have the potential for mycoplasma decontamination in *Orientia* culture due to the intrinsic resistance of *Orientia* to these classes of antibiotics [17]. However, unlike *Orientia* spp., *Rickettsia* spp. are susceptible in vitro to fluoroquinolones and to high gentamicin concentrations [18].

In the search for new antibiotic regimens for mycoplasma decontamination of rickettsial cultures, daptomycin (a lipopeptide antibiotic) is interesting because it is rapidly bactericidal against Gram-positive bacteria. Since daptomycin directly targets the cell membrane [19], it may rapidly kill mycoplasma organisms. Unlike Gram-positive bacteria, Gram-negative bacteria are intrinsically resistant to daptomycin because they reportedly do not contain the target of daptomycin [20]. In addition, porins in Gram-negative bacteria do not allow large molecules, including those of daptomycin, to pass through the outer membrane [21]. Since *Rickettsia* and *Orientia* are Gram-negative bacteria, we hypothesized that they are high-level resistant to daptomycin. Quinupristin/dalfopristin is a combination of streptogramin antibiotics targeting the 23S rRNA of most Gram-positive bacteria as well as those of certain Gram-negative bacteria. The combination of two streptogramins renders this compound synergistic and bactericidal against susceptible bacteria [22,23]. As quinupristin/dalfopristin is reportedly effective against *Mycoplasma* spp. [23,24], it may be effective for mycoplasma decontamination in cell cultures. However, the susceptibility of rickettsiae to this drug combination has never been determined.

Our study aimed to find an antimicrobial agent for mycoplasma decontamination in rickettsial cultures. We demonstrated the in vitro anti-mycoplasma activity of daptomycin against some common mycoplasma species in cell cultures. We also summarized the antimicrobial susceptibilities of common mycoplasma species to other antibiotics that may be used for mycoplasma decontamination. In addition, we demonstrated the acquisition of high-level resistance by some mycoplasma isolates after failed treatment with inadequate daptomycin concentrations or durations. Accordingly, our study provides a new option together with data for antibiotic selection in mycoplasma-contaminated *Rickettsia* and *Orientia* cultures.

## 2. Results

### 2.1. Antimicrobial Susceptibilities of Mycoplasma, Rickettsia, and Orientia

We found three species of *Mycoplasma*—four *M. orale* isolates, two *M. arginini* isolates, and one *M. hyorhinis* isolate—in our rickettsial cultures according to the DNA sequences of internal transcribed spacer (ITS) regions. The DNA sequences showed 100% similarities among these four *M. orale* isolates and two *M. arginini* isolates. Therefore, we could not conclude whether these *Mycoplasma* isolates were derived from the same or different sources.

We determined the antimicrobial susceptibilities of all of the *M. orale*, *M. arginini*, and *M. hyorhinis* isolates. Table 1 shows the results. The minimum inhibitory concentrations (MICs) of daptomycin for *M. orale* and *M. arginini* were both 2 mg/L. In contrast, *M. hyorhinis* exhibited higher resistance to daptomycin (MIC = 32 mg/L). Similar to other Gram-negative bacteria, *Rickettsia* spp. exhibited a very high-level intrinsic resistance to daptomycin (MIC > 256 mg/L). The MIC of daptomycin for *O. tsutsugamushi* (128 mg/L) was lower than that for *Rickettsia* spp., possibly due to the absence of lipopolysaccharides in the *Orientia* cell wall [25,26]. From this result, daptomycin is promising for the decontamination of *M. orale* and *M. arginini*, but it may be problematic for *M. hyorhinis* eradication, especially in *Orientia* cultures. 

Some previous studies have reported that the quinupristin/dalfopristin combination is effective for inhibiting mycoplasma, but it can inhibit certain Gram-negative bacteria such as *Legionella* and *Moraxella* [22,23]. In this study, we demonstrated that *R. typhi* (MIC = 8 mg/L) was more resistant to quinupristin/dalfopristin than *O. tsutsugamushi* (MIC = 1 mg/L). We determined the MIC of quinupristin/dalfopristin for *M. hyorhinis*, which is resistant to daptomycin. Unlike other species, including *M. pneumoniae*, *M. hominis*, and *M. fermentans* [22,23], the isolate of *M. hyorhinis* in our study was resistant to quinupristin/dalfopristin (MIC = 8 mg/L, Table 1). Since the MICs for both *Rickettsia* and *Orientia* were not very high, the agent alone is unlikely to be suitable for mycoplasma decontamination in rickettsial cultures. 

Mycoplasma organisms (except *M. pneumoniae*) are susceptible to lincosamides, including clindamycin [2,15,27,28], whereas *Rickettsia* and *Orientia* are high-level resistant to clindamycin (MIC > 32 mg/L, Table 1). Studies have reported that lincosamides can decontaminate cell cultures from mycoplasmas [16,29]. However, lincosamides are bacteriostatic, and the MBC/MIC ratio was reportedly higher than 32 [30]. Therefore, we preferred daptomycin to lincosamides in our study.

Fluoroquinolones, especially moxifloxacin, have good activity against mycoplasmas. *Rickettsia* species are susceptible in vitro, but *Orientia* is low-level resistant to fluoroquinolones [17,18] (ciprofloxacin MIC = 4, moxifloxacin MIC = 1–2 mg/L, Table 1). Therefore, fluoroquinolones cannot be used at high levels and are unlikely to be effective for mycoplasma decontamination in rickettsial cultures.

Aminoglycosides may be effective for our *M. hyorhinis* isolate, which is resistant to several antibiotics. *M. hyorhinis*, a swine pathogen, was shown to be susceptible to gentamicin in our study (MIC < 4 mg/L) as well as in other studies [31,32]. In contrast, other *Mycoplasma* spp. showed variable susceptibilities to aminoglycosides [33,34]. *Rickettsia* species are low-level resistant in vitro (MIC = 4–16 mg/L) [18], whereas *Orientia* is high-level resistant to aminoglycosides [26] (MIC > 100 mg/L, Table 1). Since our cell cultures were maintained in an antibiotic-free medium, mycoplasmas in our laboratory are likely to be susceptible to aminoglycosides. Therefore, an aminoglycoside may be effective for the decontamination of *M. hyorhinis* in *Orientia* cultures.

### 2.2. Acquisition of High-Level Resistance to Daptomycin after Incomplete Mycoplasma Eradication

At the beginning of our trial, we determined an MIC of only one isolate of *M. orale*. We treated the contaminated *Rickettsia* and *Orientia* cultures with 32 mg/L of daptomycin (half of maximum plasma concentration (*C*_max_) in patients at a therapeutic dose of 6 mg/kg [36]) for only 2 weeks. This concentration completely eradicated all *M. orale* isolates and one *M. arginini* isolate. However, another *M. arginini* isolate developed high-level resistance, and its MIC increased from 2 to 64 mg/L. The isolate of *M. hyorhinis* was inhibited but not killed by a daptomycin concentration of 32 mg/L. After treatment, its MIC increased from 32 to 256 mg/L. 

### 2.3. Successful Protocols for Complete Eradication of Mycoplasma

For mycoplasma-contaminated *Rickettsia* cultures, we recommended using a high concentration (256 mg/L) of daptomycin for 3 weeks to prevent mycoplasma relapse, although lower concentrations of 32 mg/L were effective for all *M. orale* isolates. Even though one *M. arginini* isolate in a *R. japonica* culture had acquired resistance to daptomycin (64 mg/L) after a failed treatment, we still completely eradicated this isolate by applying a high concentration (256 mg/L) of daptomycin (Table 2). Neither apparent morphological changes in L929 cells nor the retardation of *Rickettsia* growth were observed in cultures treated with high doses of daptomycin.

For mycoplasma-contaminated *Orientia* cultures, low daptomycin concentrations (32–64 mg/L) must be used instead of high doses because the MIC of daptomycin for *Orientia* is 128 mg/L. We successfully used daptomycin (32 mg/L) for 3 weeks to eradicate two isolates of *M. orale* in cultures of Karp and Kato strains. At 64 mg/L of daptomycin, *O. tsutsugamushi* still grew well for 6 days during our antimicrobial testing experiment, but we did not continue the experiment beyond that period. Aminoglycosides (gentamicin or amikacin at 100 mg/L) can be used alone or in combination with daptomycin. Although *M. hyorhinis* (in *O. tsutsugamushi* strain Gilliam culture) had intrinsic and acquired high-level resistances to daptomycin, we completely eradicated this mycoplasma isolate by treatment with either gentamicin or amikacin at 100 mg/L for 3 weeks (Table 2).

Studies have reported that lincosamides can be used for mycoplasma decontaminations of cell cultures [16,29]. We completely eliminated *M. arginini* and *M. orale* using clindamycin (32 mg/L) for 3 weeks, but failed to eliminate *M. hyorhinis* (data not shown).

We also tried to eradicate *M. hyorhinis* in L929 cultures without rickettsiae. We inoculated *M. hyorhinis* (MIC daptomycin = 32 mg/L) into sterile L929 cultures and added daptomycin 256 mg/L into these contaminated cultures. We found that daptomycin alone failed to completely eradicate *M. hyorhinis*. Since certain mycoplasma species, including *M. hyorhinis* and *M. hominis*, have been reported to survive inside eukaryotic cells [37,38,39], daptomycin alone may have failed because of its poor intracellular penetration [40]. We hypothesized that antibiotics with high intracellular concentrations, including clindamycin and quinupristin/dalfopristin, might kill intracellular mycoplasmas, and the combination with daptomycin may lead to complete eradication. Table 2 shows that the combinations of daptomycin plus either clindamycin or quinupristin/dalfopristin, and the three-drug combination did completely eradicate *M. hyorhinis.* A remarkable finding was that the quinupristin/dalfopristin concentration of 2 mg/L was the sub-MIC level that, when used alone, did not inhibit the growth of extracellular *M. hyorhinis* and intracellular *R. typhi* (Table 1). The combination of daptomycin and protein synthesis inhibitors may be applied for mycoplasma decontamination in both sterile and *Rickettsia*-infected cell cultures.

All of the effective treatment regimens in Table 2 were able to remove high numbers of mycoplasmas (approximately 0.5–1 × 10^9^ organisms per milliliter of cell culture supernatant). After 3 weeks of treatment, mycoplasmas became undetectable (<10^3^ organisms/mL). The treated samples remained mycoplasma-free for 2 consecutive weeks after treatment was discontinued.

## 3. Discussion

Mycoplasma contamination is a common and serious problem in cell cultures. Although several mycoplasma removal agents are suitable for use in sterile cell cultures, these drugs may not be applicable to cell cultures with intracellular bacteria such as rickettsiae. In this study, we demonstrated the in vitro anti-mycoplasma activities of some antibiotics against three common mycoplasma species in cell cultures. We paid attention to daptomycin because we hypothesized that it might exert rapid bactericidal action against extracellular mycoplasmas, which are Gram-positive bacteria. In addition, we demonstrated that some mycoplasma isolates developed high-level resistance to daptomycin if it was used in inadequate concentrations or for inadequate durations. We demonstrated that daptomycin or clindamycin alone completely eliminated *M. orale* and *M. arginini*, but not *M. hyorhinis*. However, the combination of daptomycin and protein synthesis inhibitors (clindamycin, quinupristin/dalfopristin, or both) did completely eradicate *M. hyorhinis*. Besides, gentamicin or amikacin alone can be used for *M. hyorhinis* contamination in *Orientia* cultures. Our new results from using antibiotics for mycoplasma elimination in rickettsial cultures, as well as data on antimicrobial susceptibilities, will be useful in work on cell cultures of intracellular organisms.

The *M. hyorhinis* isolate in this study displayed higher resistance to daptomycin than *M. orale* and *M. arginini* did. We suspect that the resistance of *M. hyorhinis* to daptomycin and quinupristin/dalfopristin is intrinsic, because daptomycin is not widely used and quinupristin/dalfopristin is still unavailable in Thailand. The susceptibilities of the three *Mycoplasma* species to daptomycin and quinupristin/dalfopristin may have differed because of cell membrane structural differences and variations in 23S rRNA sequences; therefore, further exploratory studies are necessary. Despite the resistance of *M. hyorhinis* to quinupristin/dalfopristin, we hypothesized that this drug combination may, nonetheless, be active against intracellular mycoplasmas because the intracellular concentrations of quinupristin and dalfopristin are 50 and 30 times higher, respectively, than the extracellular concentrations [23]. With clindamycin, as with quinupristin/dalfopristin, the ratio of intracellular to extracellular concentration is very high, ranging from 10 to 40 [40]. Therefore, these protein synthesis inhibitors may be effective for *M. hyorhinis* eradication if used in combination with daptomycin. 

As mentioned previously, our results demonstrated that daptomycin alone permanently eradicated *M. orale* and *M. arginini*. Even for *M. arginini* with acquired resistance (MIC = 64 mg/L), a daptomycin concentration (256 mg/L) of only four times above the MIC was enough to eradicate this *M. arginini* isolate. In contrast, *M. hyorhinis* (MIC = 32 mg/L) contamination was intractable with high doses of daptomycin alone. Previous studies have shown that certain *Mycoplasma* species, including *M. hominis* (a human pathogen) and *M. hyorhinis* (a swine pathogen), are able to invade and persist inside mammalian cells [37,38,39]. These intracellular mycoplasmas may not be inhibited by daptomycin because the intracellular concentration of daptomycin is insufficient. In contrast, clindamycin and quinupristin/dalfopristin exhibit very high intracellular concentrations and are likely to be effective against intracellular *M. hyorhinis* organisms. The quinupristin/dalfopristin concentration of 2 mg/L in this study did not inhibit the growth of extracellular *M. hyorhinis*. Clindamycin, on the other hand, could temporarily inhibit *M. hyorhinis* growth, but the organisms rapidly regrew after discontinuation of clindamycin, suggesting that clindamycin is bacteriostatic against extracellular *M. hyorhinis*. However, the combination of daptomycin plus quinupristin/dalfopristin and/or clindamycin did, as mentioned, permanently eradicate *M. hyorhinis.* We postulated that the intracellular concentrations of clindamycin and quinupristin/dalfopristin may be high enough to exceed the minimum bactericidal concentrations against *M. hyorhinis*. Collectively, these protein synthesis inhibitors are efficacious against intracellular mycoplasmas and thereby enhance the effectiveness of daptomycin, which primarily kills extracellular mycoplasmas.

Although both *Rickettsia* and *Orientia* are intrinsically resistant to daptomycin, the MIC of daptomycin for *O. tsutsugamushi* (128 mg/L) was lower than that for *Rickettsia* spp. Accordingly, daptomycin cannot be used at very high concentrations in *Orientia* culture. Nevertheless, gentamicin or amikacin can be used alone or in combination with low daptomycin concentrations because of the intrinsic high-level resistance of *Orientia* to aminoglycosides. Evidence demonstrates that aminoglycosides accumulate in phagosomes, function in non-acidic conditions, and are active against some intracellular bacteria [41]. Therefore, they are likely to be effective for both extracellular and intracellular *M. hyorhinis*.

Despite their strong in vitro susceptibility to daptomycin, some *Mycoplasma* isolates can develop high-level resistance to daptomycin if the treatment is maintained for only 2 weeks. Therefore, we recommend extended treatment to prevent relapse; in fact, we did not observe any relapse after an effective treatment for 3 weeks. The acquired resistance to daptomycin after treatment has also been observed in other bacteria, such as in staphylococci [42,43]. Further studies are required to determine the mechanism of daptomycin resistance in mycoplasmas. In addition to the optimal duration and dose of daptomycin, one or more protein synthesis inhibitors active against intracellular mycoplasma organisms should be administered to prevent relapse. Clindamycin, quinupristin/dalfopristin (low dose for *Rickettsia* cultures) or aminoglycosides (for *Orientia* cultures) show promise for use in combination with daptomycin because of their high intracellular concentrations. Since the MICs of quinupristin/dalfopristin may vary among different *Rickettsia* species, the optimal concentration should be determined before use in decontamination.

## 4. Materials and Methods

### 4.1. Antibiotics

Antibiotic powders of daptomycin (Merck & Co. Inc., Kenilworth, NJ) and quinupristin/dalfopristin (AG Scientific, San Diego, CA) were dissolved in sterile water for injection and stored as stocks at −20 °C. Sterile liquid formulations of clindamycin (Pfizer, New York, NY), moxifloxacin (Bayer AG, Leverkusen, Germany), gentamicin (GPO, Bangkok, Thailand), and amikacin (Siam Bheasach, Bangkok, Thailand) were stored at 4 °C. Final antibiotic solutions were freshly prepared before use by diluting stock solutions with cell culture medium.

### 4.2. Cell Line, Rickettsia, and Orientia Culture

The L929 mouse fibroblast cell line was obtained from Prof. Stuart Blacksell (Mahidol Oxford Tropical Medicine Research Unit) and cultivated in RPMI 1640 supplemented with 5% fetal bovine serum (Gibco, Grand Island, NY) in a humidified atmosphere with 5% CO_2_. *O. tsutsugamushi* (Karp, Kato, and Gilliam strains), *R. typhi* and spotted fever rickettsiae (*R. conorii*, *R. helvetica* and *R. japonica*)*-* infected cells were maintained at 37, 35, and 32 °C, respectively. These rickettsial organisms were obtained from Prof. Didier Raoult (Aix-Marseille Université) and Dr. Wuttikorn Rodkvamtook (Armed Forces Research Institute of Medical Sciences). Heavily infected cells were detached by cell scrapers, cryo-preserved in RPMI 1640 with 25% fetal bovine serum and 7% DMSO, and stored as rickettsia-infected cell stocks at −80 °C. For some experiments, infected cells were disrupted by repeated passage through a 25-G needle using a syringe [44]. The cell suspension was centrifuged at 400 × *g* for 5 min to discard cell pellets. The supernatants containing extracellular rickettsiae were frozen in a cryopreservative medium consisting of RPMI 1640 with 25% fetal bovine serum and 7% DMSO at −80 °C until use.

### 4.3. Mycoplasma Detection and Identification

Mycoplasma organisms were detected by real-time PCR with hydrolysis (Taqman) probes targeting the 16S rRNA gene of the genus *Mycoplasma*. We identified the *Mycoplasma* species by PCR with DNA sequencing of nuclear ribosomal internal transcribed spacer (ITS) region. Table 3 displays the sequences of primers and probes used.

### 4.4. Antimicrobial Susceptibility Testing

Mycoplasma organisms were cultivated in cell culture media containing L929 monolayers in order to obtain a large number of organisms in the log phase. We extracted DNA from supernatants containing mycoplasmas using the QIAamp DNA mini kit (Qiagen, Hilden, Germanyand). We used real-time PCR to quantify the copy number of mycoplasmas using the standard curve method. To determine the MICs, approximately 5 × 10^5^ mycoplasma organisms per ml were inoculated onto L929 monolayers at a multiplicity of infection of 2:1. The contaminated cells were then grown in RPMI 1640 medium with 5% fetal bovine serum with different antibiotic concentrations. We quantified extracellular mycoplasma DNA copy numbers in cell culture supernatants on days 0, 1 and 2 by real-time PCR and compared them to that of time 0. The mycoplasma count at the MIC was confirmed again on day 3. We chose real-time PCR because it yields rapid results and represents the real culture conditions of rickettsiae with 5% serum.

Antimicrobial testing of *O. tsutsugamushi* and *Rickettsia* spp. was performed as previously described [17,35]. Briefly, extracellular rickettsiae were pre-incubated with indicated antibiotics for 15 min, inoculated onto L929 cells, and further incubated at 37 °C for 1 hour. Afterwards, infected cells were washed and maintained in media with different concentrations of antibiotics. The intracellular rickettsiae on days 0, 3, and 6 were quantitated by real-time PCR using primers and probes shown in Table 3.

### 4.5. Mycoplasma Decontamination of Rickettsia Cultures

We pre-incubated mycoplasma-contaminated rickettsiae with the indicated antibiotics and used them to infect mycoplasma-free L929 cells. We manipulated only one mycoplasma-contaminated culture at a time to prevent cross-contamination from another mycoplasma isolate. We trypsinized the infected cells and transferred them to new flasks every 3–4 days. Rickettsiae from infected cells with cytopathic effect were passaged into new cells, as appropriate. We grew the infected cells in media with antibiotics for 3 weeks and subsequently in media without antibiotics for 2 weeks. Mycoplasma testing was performed every week after the discontinuation of antibiotics.

## 5. Conclusions

In conclusion, our results demonstrated that daptomycin is suitable for complete eradication of *M. orale* and *M. arginini* in *Rickettsia* and *Orientia* cultures. For daptomycin-resistant *M. hyorhinis* in *Orientia* cultures, an aminoglycoside can be used instead of daptomycin. Moreover, the combinations of daptomycin plus protein synthesis inhibitors (clindamycin and/or quinupristin/dalfopristin) were demonstrated to be effective in eradicating *M. hyorhinis* in sterile cell cultures. A treatment duration of at least 3 weeks is recommended to prevent relapse. Our mycoplasma decontamination methods with antibiotics are more convenient and practical than other methods involving in vivo passage in mice, which requires a longer duration [45] and animal biosafety level 3 laboratories [46]. Further studies with larger numbers of isolates and more species of *Mycoplasma* are needed to determine the effectiveness and consistency of our protocols.

## Figures and Tables

**Table 1 antibiotics-08-00123-t001:** MICs (mg/L) of Different Antibiotics for *Mycoplasma* spp., *Rickettsia* spp., and *Orientia tsutsugamushi.*

Drugs	MICs (mg/L) for organisms		
	*Mycoplasma* spp.	*Rickettsia* spp.	*O. tsutsugamushi*
Daptomycin	2 (*M. orale*, *M. arginini*) 32 (*M. hyorhinis*) *	>256 (*R. typhi*, *R. japonica*, *R. helvetica*) *	128 (Karp) *
Quinupristin/Dalfopristin	0.05–2 [23,24] 8 (*M. hyorhinis*) *	8 (*R. typhi*) *	1 (Gilliam) *
Clindamycin	Susceptible ≤1 * [27,28,30]	>32 *	>32 *
Fluoroquinolones	Susceptible ≤0.12(MXF) [24,28,30]	0.25–1 (CIP) [18,35]	4 (CIP) [17] 2 (MXF for Karp) 1 (MXF for Kato, Gilliam) *
Aminoglycosides (gentamicin)	<4 (GEN, *M. hyorhinis*) *, [31,32] <0.25–10 (GEN, *M. hominis* and *M. fermentans*) [33,34]	4–16 [18]	>100 (GEN, AMK) *

* Data from this study. The superscript numbers indicate the references. Abbreviations: MXF, moxifloxacin; CIP, ciprofloxacin; GEN, gentamicin; AMK, amikacin.

**Table 2 antibiotics-08-00123-t002:** Successful Mycoplasma Decontamination Protocols in this Study.

Cultures	Contaminants	Treatment ^1^
*R. typhi*	*M. orale*	Daptomycin 32 mg/L
*R. conorii*	*M. arginini*	Daptomycin 32 mg/L
*R. helvetica*	*M. orale*	Daptomycin 32 mg/L
*R. japonica*	Mixed *M. orale* and *M. arginini*	Daptomycin 32 mg/L for *M. orale* followed by Daptomycin 256 mg/L for acquired resistant *M. arginini* (MIC = 64 mg/L)
*O. tsutsugamushi* Kato	*M. orale*	Daptomycin 32 mg/L ^2^
*O. tsutsugamushi* Gilliam	*M. hyorhinis*	Gentamicin 50–100 mg/L or Amikacin 100 mg/L
L929 cells without rickettisae	*M. hyorhinis* (experimental contamination)	Daptomycin 256 mg/L plus either clindamycin 32 mg/L or quinupristin/dalfopristin 2 mg/L, or 3-drug combination

^1^ Duration of treatment for 3 weeks. ^2^ Can be combined with an aminoglycoside.

**Table 3 antibiotics-08-00123-t003:** Primer and Probe Sequences.

Primer or Probe	Sequences 5’-->3’
Primers	
Mycop 16S F	GGA GCT GGT AAT RCC CAA AGT C
Mycop 16S R	CCA TCC CCA CGT TCT CGT AG
OT 47-kDa F	CCA TCT AAT ACT GTA CTT GAA GCA GTT GA
OT 47-kDa R^1^	GTC CTA AAT TCT CAT TTA ATT CTG GAG T
TG *ompB* F	GTG CAG TAT CTT CAG GTG ATG A
SFG *ompB* F	GGT GAC GAG GCT GTT GAY AAT G
TG/SFG *ompB* R	GGY IGT TTT TGC TTT ATA ACC AGC TA
Mycop ITS F	CCT AAG GYA GGA CTG GTG ACT GG
Mycop ITS R	CAC GTC CTT CWT CGA CTT TCA GAC
(sequencing)	
Probes	
Mycop 16S R^a^	FAM-CCC AGT CAC CAG TCC TGC CTT AGG-BHQ1
OT 47-kDa R^b1^	FAM-TCA TTA AGC/ZEN/ATA ACA TTT AAC ATA CCA CGA CGA-IBFQ
TG *ompB* R^b^	MAX-TTC TGC GAT GTT ATA GAA AGG TTT AGC CCA- BHQ1
SFG *ompB* R^c^	Texas red-ATG TGC ATC AGT ATA GAA AGG TTT TGC CC-BHQ2

^a^^b^^c^ purchased from Biodesign (Bangkok, Thailand), Integrated DNA Technologies (Coralville, IA), and Eurogentec (Seraing, Belgium), respectively. ^1^ Reference [44].

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
