# Peer review of "Anti-Mycoplasma Activity of Daptomycin and Its Use for Mycoplasma Elimination in Cell Cultures of Rickettsiae"

_antibiotics, 2019, doi:10.3390/antibiotics8030123_

Round 1
Reviewer 1 Report
The authors "aimed to find an antimicrobial agent for mycoplasma decontamination in rickettsial cultures". Although the experimental data can be interesting for scientists working in this specific laboratory setting, actually the presentation is quite unusual. The fusion between results and discussion is not a good solution, especially in order to appreciate the general findings of the research and the message of the article. Moreover, especially in this kind of articles, material and methods should be described before the results.
Author Response
Responses to the comments of Reviewer #1
Thank you for your review of our paper. We have answered each of your points below.
The authors "aimed to find an antimicrobial agent for mycoplasma decontamination in rickettsial cultures". Although the experimental data can be interesting for scientists working in this specific laboratory setting, actually the presentation is quite unusual. The fusion between results and discussion is not a good solution, especially in order to appreciate the general findings of the research and the message of the article. Moreover, especially in this kind of articles, material and methods should be described before the results.
Response: As you suggested, we separated the Discussion section from the Results in the revised manuscript. However, we still placed the Materials and Methods after the Discussion because it is the format of the journal of MDPI publisher.
Moreover, while waiting for the decision, we have performed more experiments using the combinations of daptomycin plus clindamycin, daptomycin plus quinupristin/dalfopristin, and 3-drug combination. All of these combinations could eradicate daptomycin-resistant Mycoplasma hyorihinis. The results were described in results on line 153-166, Table 3 and the Discussion part.
Reviewer 2 Report
The manuscript by Tantibhedhyangkul and colleagues describes the use of daptomycin to eliminate various mycoplasma species from contaminated cell culture. Although a few additional details are required, this may prove useful for those seeking to rescue critical cultures that have become mycoplasma infected. In addition, this may be the first report of acquired resistance to daptomycin in certain mycoplasma lineages.
Scientific Points
1. There are a number of statements of fact that are not referenced. These include L34, L38, L39
2. L42. Some Gram-negative organisms are susceptible to clindamycin. The authors should consider revising this statement to “.. have no effect on many gram-negative bacteria.”
3. L54. This sentence may not read quite what the authors intended. Presumably, the outer membrane provides a barrier and that the porins embedded within this may not allow uptake of the drug.
4. Section 2.1 and the Methods section. It is not specifically stated how these mycoplasma isolates were obtained. Are they all derived from contaminated cultures? How were they purified to single colony isolates? Were they filter cloned? Were the 4 isolates of M. orale possibly identical (i.e introduced by the same source). Without the origin, purification etc, the reader cannot evaluate whether 1 strain or 4 are being tested. Were their SNPs in the ITS region sequenced (that might be informative for this question)?
5. L76 The authors state that they have measured the MIC for decontaminating cell culture. The methods (L192) indicate that they were grown in L929 culture media. This is ambiguous, as written. Were the cells grown in media alone (i.e that used for growing L929 cells) or were they grown in cell culture with cells? In which case, what was the MOI etc? If L929 cells were not included, then the authors were not measuring the MIC for decontamination, but just the MIC.
6. L80. The authors might use “possibly” rather than “probably” unless data are available that show the influence of LPS structure on daptomycin uptake (if so, then this should be cited).
7. L125. Since the authors are working with contaminated cultures, do the authors have evidence that no additional contamination occurred over the 2 week decontamination procedure? i.e. the resistant strain has arisen during treatment rather than being introduced?
8. L139 and L139. The authors state that there was no toxicity and no effect on Rickettsia growth. As these are “data not shown” more details about how these were assessed should be provided. If they were not directly tested, the authors might state that there was no apparent (or no observable)….
9. L155. This is overstated. I would recommend that the authors raise this as a possibility rather than “probably due”. The authors do not present any data that indicate that the isolates of the three species tested had differing cytadherence to the L929 target cells.
10. Table 4. References for the primers (if previously used and validated), should be provided in the Table.
11. The text (results or discussion sections) should include the number of organisms that were removed by this method and what the limit of detection was for the PCR assay.
12. The authors might consider including a reference (or two) about the extent of mycoplasma contamination in tissue cultures worldwide, since this is a significant problem.
Language points/suggestions
1. Throughout most of the manuscript, genus and species names are not italicized.
2. L26, L33 and elsewhere, depending on journal style, in vitro may require italicization.
3. L43 “are” does not match “class”. I suggest using…..contains bacteriostatic agents…
4. L43. “permanently killed” should be re-worded as dead is dead.
5. L56 A better transition would improve the flow between these sentences.
6. L57 and elsewhere (L188): 23S with Svedberg unit capitalized (some journals may require italicization also for this unit.
7. L61 for this drug combination has never…
8. L72. I recommend avoiding 1 sentence paragraphs.
9. L77. Similar to other gram-negative….
10. L92 Please insert a space after “Table”
11. L95 The authors have already commented on the intrinsic resistance for Mhyo on L83
12. Table 1. The abbreviations MXF, GEN, CIP, AMK should be defined as footnotes
13. L128. I recommend removing the word “this”, as the precise mechanism has not been established.
14. L136 in a R. japonica culture
15. Table 3 footnotes; numerals should be superscripts.
16. L160 can be used
17. L168 check the spacing for Merck&Co
18. L175, L180, L223 and L224. Add spaces after the abbreviated title and the names of the scientists. Also for L180 and L224, depending on journal style, including 2 titles might not be necessary (for Didier Raoult). Also, L180 and L224, the university name does not require abbreviation.
19. L188 We identified the Mycoplasma …
Reviewer 3 Report
This study by Tantibhedhyangkul et al. Contains manuscript new information and protocols that will be useful for those working with Rickettsiales. It is well written but does require a few minor edits.
Line 17 and elsewhere. The authors need to italicize all scientific names. In some sections they are italicized.
Line 22. Be more specific and give the aminoglycoside in parentheses (gentamicin).
Lines 83 and 97. Are the authors referring to Thailand? If so, use Thailand. Or state …..in our country.
Table 1. State specifically which aminoglycoside(s) and fluoroquinolone(s) you are referring to.
Line 140. Do you mean Orientia?
Line 148. Remove the s from isolates.
Line 169. Injection? Do you mean inoculation or addition?
Line 185. State composition of cryopreservation medium.
Line 195. Figure S1 is superfluous. Please delete.
Author Response
Responses to the comments of Reviewer #3
Thank you for your review of our paper.
In this file, we have responded to your comments.
Moreover, while waiting for the decision, we have performed more experiments using the combinations of daptomycin plus clindamycin, daptomycin plus quinupristin/dalfopristin, and 3-drug combination. All of these combinations could eradicate daptomycin-resistant Mycoplasma hyorihinis. The results were described in results line 153-166, Table 3 and the Discussion part.
We have answered each of your points below.
Comments and Suggestions for Authors
This study by Tantibhedhyangkul et al. Contains manuscript new information and protocols that will be useful for those working with Rickettsiales. It is well written but does require a few minor edits.
Line 17 and elsewhere. The authors need to italicize all scientific names. In some sections they are italicized.
Response : These mistakes may occur after the manuscript reformatting by the editorial office because all scientific names were italicized in the original manuscript. However, we have corrected these mistakes in this revised version.
Line 22. Be more specific and give the aminoglycoside in parentheses (gentamicin).
Response : Correction on line 21 ‘An aminoglycoside (gentamicin)’
Lines 83 and 97. Are the authors referring to Thailand? If so, use Thailand. Or state …..in our country.
Response : Correction in the discussion on line 187-189 ‘We suspect that the resistance of M. hyorhinis to daptomycin and quinupristin/dalfopristin is intrinsic because daptomycin is not widely used and quinupristin/dalfopristin is still unavailable in Thailand.’
Table 1. State specifically which aminoglycoside(s) and fluoroquinolone(s) you are referring to.
Response : We indicated (gentamicin) below aminoglycosides in the Table 1. However, mycoplasmas are well susceptible to several agents in fluoroquinolone class including ciprofloxacin, levofloxacin, moxifloxacine, etc. Therefore, we did not indicate the specific agent(s) in the left column below fluoroquinolones. However, we indicated the name(s) in abbreviations for each of quinolones tested for MICs.
Line 140. Do you mean Orientia?
Response : Correction on line 138 ‘For mycoplasma-contaminated Orientia cultures’
Line 148. Remove the s from isolates.
Response : Correction on line 145-146 ‘this mycoplasma isolate’
Line 169. Injection? Do you mean inoculation or addition?
Response : Actually the term ‘injectable formulation’ meant ‘Sterile liquid formulations for parenteral injection’. However, for clear understanding, we used the term ‘Sterile liquid formulations’ on line 252 in this revised manuscript.
Line 185. State composition of cryopreservation medium.
Response : Correction on Line 268-269 ‘cryopreservative medium consisting of RPMI 1640 with 25% fetal bovine serum and 7% DMSO’.
Line 195. Figure S1 is superfluous. Please delete.
Response : We deleted Fig. S1 as you suggested.
Round 2
Reviewer 1 Report
The authors improved a lot their manuscript and, importantly, they made important changes in the manuscript structure (e.g. separation between results and discussion), as suggested. This essential improvement made the article much more appealing for the reader, in my opinion.
Therefore, considering the scientific value of these experimental study, I have some minor concerns/suggestions below to be addressed in order to make this manuscript suitable for publication on the journal.
INTRODUCTION:
-even though this paper is focused on Mycoplasma spp. as cell culture contaminants in the laboratory, I recommend the authors to spend a few sentence at the very beginning to highlight how some Mycoplasma spp. are very important human pathogens as well, since they can cause several and different diseases (refer to: Curr Opin Rheumatol. 2018 Jul;30(4):380-387. doi: 10.1097/BOR.0000000000000494. Review; J Infect Dis. 2017 Jul 15;216(suppl_2):S420-S426. doi: 10.1093/infdis/jix200. Review)
RESULTS:
-Table 2 should be removed, as it displays very few data which are appropriately summarized in the text.
Author Response
We followed your suggestions.
In this revised version, We described about pathogenic and non-pathogenic mycoplasmas in the paragraph 2 of introduction.
We also deleted table 2 from the manuscript.
Best regards